# Unhealthy Food Consumption Is Associated with Post-Acute Sequelae of COVID-19 in Brazilian Elderly People

**DOI:** 10.3390/idr17020025

**Published:** 2025-03-13

**Authors:** Guilherme José Silva Ribeiro, Rafaela Nogueira Gomes de Morais, Olufemi Gabriel Abimbola, Nalva de Paula Dias, Mariana De Santis Filgueiras, André de Araújo Pinto, Juliana Farias de Novaes

**Affiliations:** 1Department of Nutrition and Health, Universidade Federal de Viçosa (UFV), Viçosa 36570-900, MG, Brazil; rafaela.morais@ufv.br (R.N.G.d.M.); olufemi.abimbola@ufv.br (O.G.A.); nalva.dias@ufv.br (N.d.P.D.); mariana.filgueiras@ufv.br (M.D.S.F.); jnovaes@ufv.br (J.F.d.N.); 2Health Sciences Center, Universidade Estadual de Roraima (UERR), Boa Vista 69306-530, RR, Brazil; andre.pinto@uerr.edu.br

**Keywords:** elderly, food consumption, Long COVID, nutritional epidemiology, infection

## Abstract

**Background/Objectives**: The factors associated with post-acute sequelae of COVID-19 (PASC) are not yet fully understood in developing countries. Our objective was to investigate the relationship between food consumption and the occurrence of PASC in Brazilian elderly people. **Methods**: This cross-sectional study included 1322 elderly people aged 60 or over, infected with SARS-CoV-2 in 2020, living in the state of Roraima in Brazil. Using the Brazilian National Food and Nutrition Surveillance System (SISVAN, in Portuguese) tool, food consumption markers were evaluated. The persistence of post-acute sequelae of COVID-19 was assessed three months after SARS-CoV-2 infection. Poisson regression with robust variance was performed to estimate the prevalence ratio (PR) with a 95% confidence interval (95% CI). **Results**: Fruit consumption [PR 0.92; 95% CI: 0.85–0.99] was associated with a lower occurrence of PASC, with a significant interaction in individuals aged 60 to 69 years old, not hospitalized, and those without chronic kidney disease. In addition, the consumption of sugar-sweetened beverages [PR 1.23; 95% CI: 1.12–1.35], sandwich cookies, sweets, and treats [PR 1.12; 95% CI 1.03–1.22] was positively associated with the occurrence of PASC in the elderly people, with a significant interaction in individuals living in the capital and without hypercholesterolemia. **Conclusions**: Unhealthy food consumption was associated with PASC in Brazilian elderly people. An improvement in the diet quality of elderly people is necessary to minimize health complications in PASC.

## 1. Introduction

The COVID-19 pandemic, caused by SARS-CoV-2, was not only associated with acute illness but also with long-term health complications [1]. It is estimated that approximately 65 million people worldwide are impacted by lasting symptoms even after their recovery [2]. The so-called Post-Acute Sequelae of COVID-19 (PASC) or “long COVID”, refers to persistent symptoms and complications after the acute phase of the disease [3]. This new and complex condition has increased significantly throughout the world [4,5] and in Brazil, specifically, it has caused significant public health problems [6].

PASC can result in long-term pulmonary, cardiovascular, neurological, psychiatric, and digestive symptoms lasting between two or three months, not being explained by other diagnoses [2,5,7]. It is believed that, out of every 10 survivors, one has persistent symptoms for more than 12 weeks, regardless of the severity of the infection or previous hospitalization [8]. Persistent symptoms include fatigue, shortness of breath, palpitations, chest pain, headache, anxiety, depression, and hair loss, among others [4,7,9].

There is evidence that sociodemographic characteristics (female sex and older age), pre-existing medical conditions such as hypertension and diabetes [10], and a lack of vaccination against COVID-19 [3] are associated with PASC. Lifestyle factors, such as a sedentary lifestyle, smoking, alcohol, and an inadequate diet, contribute to increasing the prevalence and severity of PASC in adults [11,12,13].

Diet quality has been widely recognized as part of the treatment of SARS-CoV-2 viral infection [3]. Maintaining a healthy diet, including fruits and vegetables, can effectively prevent and reduce the risk of severe COVID-19 [14] as the vitamins, minerals, and antioxidants of these foods can strengthen the immune system, reduce inflammation, and help in recovery from COVID-19 [15,16]. On the other hand, poor diet quality can affect nutritional status, increasing the risk of COVID-19 infection and severity, making recovery difficult [17,18]. However, few studies have investigated the relationship between food consumption in recovery from COVID-19 in elderly people. In this sense, understanding this relationship is crucial for healthcare planning and intervention strategies in developing countries.

Therefore, we aimed to investigate the associations of food consumption with the occurrence of PASC in Brazilian elderly people. Our hypothesis is that the consumption of unhealthy foods increases the occurrence of PASC in the Brazilian elderly group.

## 2. Materials and Methods

### 2.1. Design and Location

This cross-sectional study was conducted based on health-monitoring records of elderly people in the state of Roraima, Brazil. Information from elderly patients (people aged 60 or over) treated in primary healthcare between January and December 2020 was included. The data were recorded by the state’s Department of Epidemiological Surveillance (*Departamento de Vigilância Epidemiológica*, DVE in Portuguese), collected by health professionals, and entered into a control system maintained by the Roraima State Health Department. Access to the data was granted in 2022 after obtaining the necessary authorizations from the competent regulatory bodies (State Health Department of Roraima). The study received approval from the Human Research Ethics Committee of the State University of Roraima (opinion no. 5385012), in accordance with the guidelines of the National Health Council.

The state of Roraima, located in the Western Amazon region in the extreme north of Brazil, has 15 municipalities and a territorial area of approximately 223,644.53 km^2^. Roraima has a total population of 636,303 inhabitants, 16% of whom are 60 years old or over, and a demographic density of approximately 2.85 inhabitants per km^2^. The capital, Boa Vista, has an estimated population of 413,486 inhabitants, representing around 65% of the state’s total population.

### 2.2. Sample Selection and Data Collection

This study selected records of elderly people diagnosed with SARS-CoV-2 through nasal swabs confirmed by polymerase chain reaction throughout 2020. Individuals with complete information and who received follow-up from healthcare professionals during the first year of the pandemic were included. The first reported case of COVID-19 in the state of Roraima occurred on 21 March 2020.

According to data from the DVE, a total of 4194 elderly people were diagnosed with SARS-CoV-2 in 2020. During the exploratory data analysis, it was found that a significant amount of information had not been entered into the internal control system of the Department of Health. Given the lack of data on PASC in elderly Brazilians, a pilot study was carried out to explore this reality and serve as a basis for the sample calculation. Using the same monitoring instrument from the Ministry of Health, 104 elderly people (not included in the final analysis) responded to a questionnaire about PASC, of which 60% reported the presence of at least one symptom lasting more than three months.

Based on the total number of elderly people diagnosed with SARS-CoV-2 in 2020 and the prevalence of symptoms reported in the pilot study, the following parameters were used to estimate the sample size: a 95% confidence level, a prevalence of 50% (for the evaluation of multiple outcomes), a tolerable error of 4%, and a correction factor for the design effect (deff) set at 2. To compensate for potential losses and ensure an adequate sample size, an additional 20% was added to the final sample number. Therefore, a minimum of 1260 participants was determined based on these sampling parameters.

In 2020, elderly people diagnosed with SARS-CoV-2 were followed up by a team of healthcare professionals via phone calls in the first few weeks after diagnosis and again three months later. Using a standardized form throughout primary healthcare in the state, the presence or absence of symptoms was reported by the elderly person themselves or their family members. During telephone calls, information was recorded in written form and subsequently entered the virtual internal control system, with the initial data collection carried out by the DVE.

## 3. Study Variables

### 3.1. Post-Acute Sequelae of COVID-19

The dependent variable in this study was the persistence of PASC—a set of symptoms and complications that persist after the acute phase of the disease [3]. PASC was assessed during the last contact made three months after SARS-CoV-2 infection [7]. Patients were asked to indicate whether they had specific symptoms, including chest pain, chest tightness, palpitations, dyspnea (shortness of breath), and leg swelling. Responses were recorded as “yes” for the presence or “no” for the absence of each symptom. Elderly people who reported at least one of these symptoms at the end of the follow-up period were considered to have PASC. For this, a score was created, assigning a value of “1” for the presence and “0” for the absence of each symptom, with a total score ranging from zero to five points. Thus, a patient was considered to have PASC if they had any score greater than or equal to one.

### 3.2. Food Consumption Markers

Food consumption was assessed using the food consumption marker form of the Brazilian National Food and Nutrition Surveillance System (*Sistema de Vigilância Alimentar e Nutricional*—SISVAN, in Portuguese), based on a quick and simplified screening questionnaire to assess the population’s diet in all basic health units in the country [19]. The protocol contains single items to assess ultra-processed foods (UPF) consumption on the previous day. For each food marker, a healthcare professional (usually a nurse) asks, “Did you eat this yesterday?” The markers are: Beans; Fresh fruit (do not consider fruit juice); Vegetables (do not include potatoes, cassava, and yam); Hamburger and/or sausages (ham, mortadella, salami, *linguiça*, and sausage); Sugar-sweetened beverages (soda, canned juice, powdered juice, canned coconut water, *guarana*/cherry syrup, fruit juice with added sugar); Instant noodles, packaged snacks or cookies; and Sandwich cookies, sweets, or treats (candies, lollipops, gum, caramels, and gelatin). The answer options were “yes”, “no”, and “I don’t know”. No responses were recorded with the option “I don’t know”. These groups were selected for their ability to reflect diet quality and health impact, based on nutritional guidelines and scientific evidence [19].

### 3.3. Covariates

The sociodemographic data collected were sex, age, education level (no study, up to 8 years, and more than 8 years), skin color/race, place of residence, and pre-existing diseases: high blood pressure, diabetes *mellitus*, hypercholesterolemia, obesity, and chronic kidney disease. Smoking and COVID-19 hospitalization were also available. All information was recorded in binary format (yes or no).

## 4. Statistical Analysis

Only complete data available were analyzed. Descriptive (absolute and relative frequency) and inferential analyses were used. Pearson’s chi-square test for dichotomous variables and chi-square test for trend for ordinal variables were used to evaluate the association between the variables. Poisson regression with robust variance was performed, adjusted for covariates (age, sex, education, level of education, location, hospitalization, hypertension, diabetes *mellitus*, hypercholesterolemia, obesity, chronic kidney disease, and smoking) to identify whether food consumption was associated with the prevalence of PASC in the elderly people. As a measure of association, the prevalence ratio (PR) and their respective 95% confidence intervals (95% CIs) were estimated. Poisson regression was chosen because odds ratios estimated by logistic regression are often overestimated in cross-sectional studies, particularly when the outcome has a high prevalence (>10.0%) [20] This is especially relevant in this study, where the prevalence of PASC among the elderly was 61.7% [21]. The interactions between food groups and covariates associated with PASC were tested by including an interaction term in the adjusted Poisson regression model. For data analysis, STATA software version 14 (StataCorp LP, College Station, TX, USA) was used. A statistical significance level of 5% was considered.

## 5. Results

The elderly people with sequelae were mainly male (64.2%), aged 80 years old or more (46.6%), indigenous (75.7%), without an education (67%), and residents in the non-metropolitan area (73.1%) (Table 1).

Fruit consumption [PR 0.92; 95% CI: 0.85–0.99] was associated with lower occurrence of PASC, and there was a significant interaction in individuals aged 60 to 69 years old, not hospitalized, and those without chronic kidney disease (Table 2, Figure 1). No association was found between the consumption of beans and vegetables with PASC in elderly people.

The consumption of sugar-sweetened beverages [PR 1.23; 95% CI: 1.12–1.35], sandwich cookies, sweets, and treats [PR 1.12; 95% CI 1.03–1.22] was positively associated with the occurrence of post-acute sequelae of COVID-19 in elderly people. There was a significant interaction among individuals living in the capital (Table 3, Figure 2). In addition, the consumption of sandwich cookies, sweets, and treats increased the probability of 12% (95% CI: 1.03–1.22) for the PASC, when compared to those who did not consume these foods. There was a significant interaction between individuals living in the capital and those without hypercholesterolemia (Figure 2). No association was found between the consumption of hamburgers and/or sausages, instant noodles, packaged snacks, or savory biscuits with the PASC in elderly people.

## 6. Discussion

In this cross-sectional study with elderly people, fruit consumption was associated with a lower likelihood of developing PASC with a significant interaction in individuals aged 60 to 69 years old, not hospitalized, and without chronic kidney disease. The opposite was observed for the consumption of sugar-sweetened beverages and sandwich cookies, sweets, and treats, which increased the likelihood of developing this condition. This association was more pronounced in individuals living in the capital and without hypercholesterolemia. To our knowledge, this is the first study to investigate this relationship in Brazilian elderly people.

We observed that fruit consumption was inversely associated with PASC in elderly people. A cohort study with approximately 68,000 elderly participants showed a similar result [22]. The oxidative stress and inflammation induced by SARS-CoV-2 infection could be mitigated by high fruit intake, a rich source of antioxidants [23], which can protect the lungs and heart against oxidative damage and hyperinflammation, in addition to preventing the severe form of COVID-19 [24,25,26,27]. Other studies found that elderly people who consumed fruit had lower levels of inflammatory markers and lower susceptibility to multisystemic sequelae [3,28]. The daily consumption of vitamins, minerals, and fiber improves immune function, protects against viral infections, and reduces the risk of SARS-CoV-2 infection [29,30,31].

In our study, the interaction analyses showed that fruit consumption was inversely associated with PASC in elderly people aged 60–69 years and who were not hospitalized. Moreover, a previous prospective study conducted with older adults revealed that a healthy lifestyle, involving behaviors such as a proper diet, weight management, and sufficient sleep, when combined with a lower number of comorbidities prior to SARS-CoV-2 infection, resulted in a lower risk of hospitalization and PASC among older individuals [11]. These findings support our hypothesis that maintaining good health conditions and a healthy lifestyle contributes as protective factors against COVID-19 complications in older adults.

We also observed that the interaction analyses demonstrated that fruit consumption was inversely associated with PASC in elderly individuals without chronic kidney disease (CKD). It is known that patients with CKD have greater comorbidities, an impaired immune system, and high levels of viral infections, in addition to low fruit consumption, compared to patients without CKD [32,33]. Therefore, this chronic disease could increase the risk of severe COVID-19 in these patients [32].

Inadequate diet appears to influence the increase in PASC in patients with COVID-19 [13]. In our study, the consumption of sugar-sweetened beverages and sandwich cookies, sweets, and treats increased the probability of PASC occurrence in the elderly. The high consumption of UPF compromises immunity and increases the risk of COVID-19 infection [34]. These foods, with an excess of food additives such as dyes, emulsifiers, stabilizers, and flavorings [35], could trigger inflammatory markers, such as interleukin IL-6, IL-15, tumor necrosis factor α (TNF α), leptin, C-reactive protein [36], and intestinal inflammation [37], which could increase the severity of COVID-19 [3]. Although these additives are approved for use in the food industry, evidence suggests that prolonged exposure may contribute to low-grade systemic inflammation, raising concerns about their long-term health effects, especially in individuals with pre-existing conditions. This is significantly relevant, as these inflammatory markers have been observed in patients with cardiopulmonary sequelae [9,38,39,40].

We found that the consumption of sugar-sweetened beverages, sandwich cookies, sweets, and candies was associated with a higher occurrence of PASC in elderly individuals living in the state capital. These findings may be related to urbanization, which causes changes in lifestyle, with greater availability of UPF. The lower income and education levels of older adults may worsen this situation because living in neighborhoods with a low supply of healthy foods often forces them to purchase more affordable but less healthy foods. The combination of limited availability and financial constraints leads to greater consumption of unhealthy foods [41,42,43].

Finally, we observed an association of consumption of sandwich cookies, sweets, and treats with PASC in elderly people without hypercholesterolemia. This finding can be attributed to the cross-sectional design, leading to reverse causality bias, once elderly people with hypercholesterolemia can take greater care of their general health, including diet. This result was identified in previous studies carried out in Brazil [44,45] and Italy [46]. Future longitudinal investigations would improve the understanding of these relationships.

Our study has some strengths, including the sample size that achieved a large number of elderly people (1322) from a little-explored region in Brazil. Second, according to our knowledge, this study appears to be the first to investigate the association between food consumption and PASC in developing countries, considering the statistical analyses adjusted for covariates. As a limitation, the method used to assess food consumption could not reflect the usual intake once it was evaluated for only one day; and it evaluated if the participant consumed the foods (yes/no), without considering the frequency of consumption and/or the portion size. However, the instrument appears to demonstrate the potential to reflect the overall quality of adults’ diets, as evidenced by previous studies [47,48]. Despite the post-COVID-19 symptoms reported by participants, it is not possible to consider that all symptoms were consequences of COVID since they are nonspecific and may be related to other pre-existing illnesses.

## 7. Conclusions

We conclude that fruit consumption was positively associated with the low occurrence of PASC, while the consumption of sugar-sweetened beverages and sandwich cookies, sweets, and treats was positively associated with its higher occurrence in Brazilian elderly people. The implementation of public health to improve the diet quality in elderly people is necessary to reduce the negative impacts of infectious diseases, such as COVID-19.

## Figures and Tables

**Figure 1 idr-17-00025-f001:**
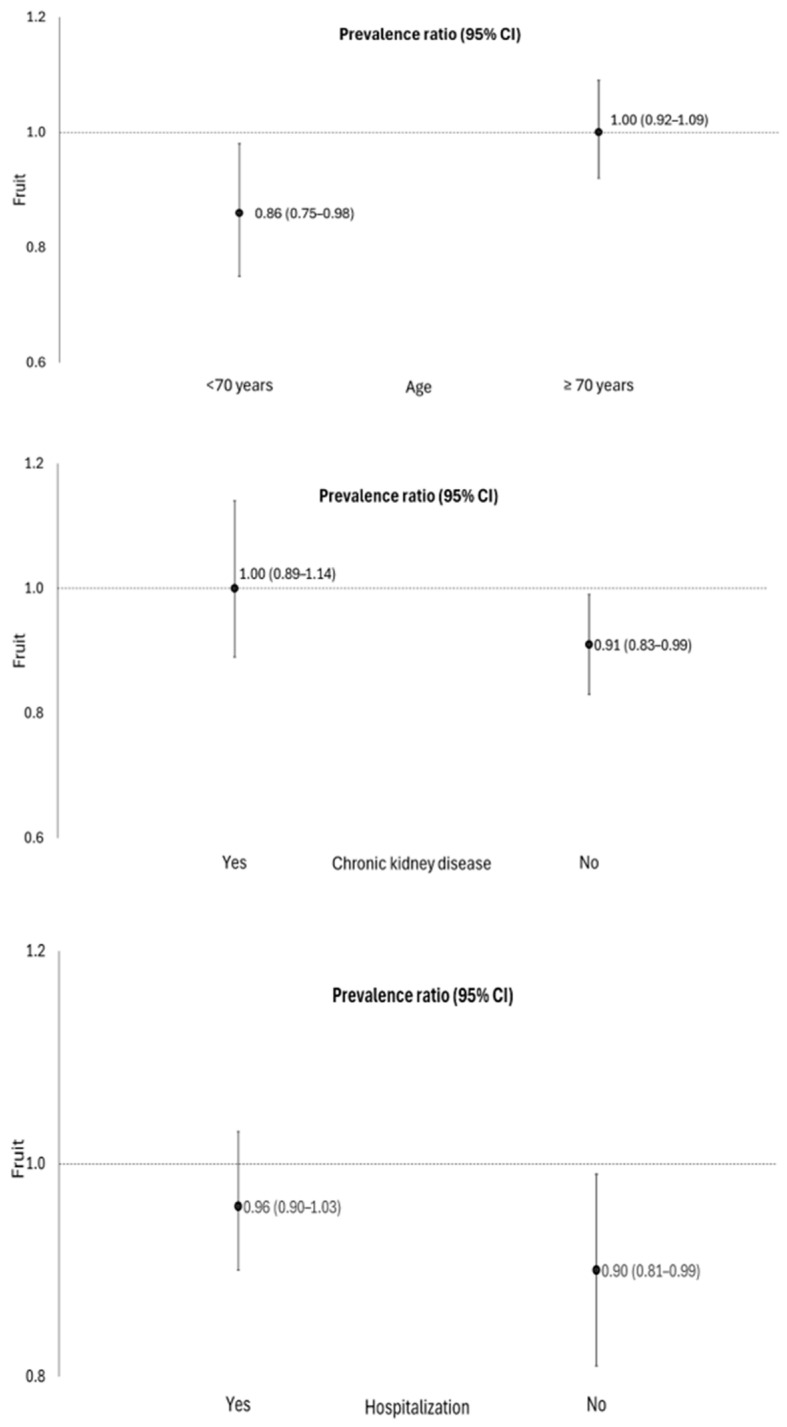
Association between fruit consumption and the occurrence of COVID-19 sequelae, stratified by age, hospitalization, and the presence of chronic kidney disease in elderly people. Roraima, Brazil, 2020. Poisson regression with robust variance, *p* < 0.05. Adjustment for sex, age, place of residence, education level, skin color/race, hospitalization, smoking, hypertension, diabetes *mellitus*, hypercholesterolemia, obesity, and chronic kidney disease. Age was categorized according to the 50th percentile of the sample, which corresponds to 70 years.

**Figure 2 idr-17-00025-f002:**
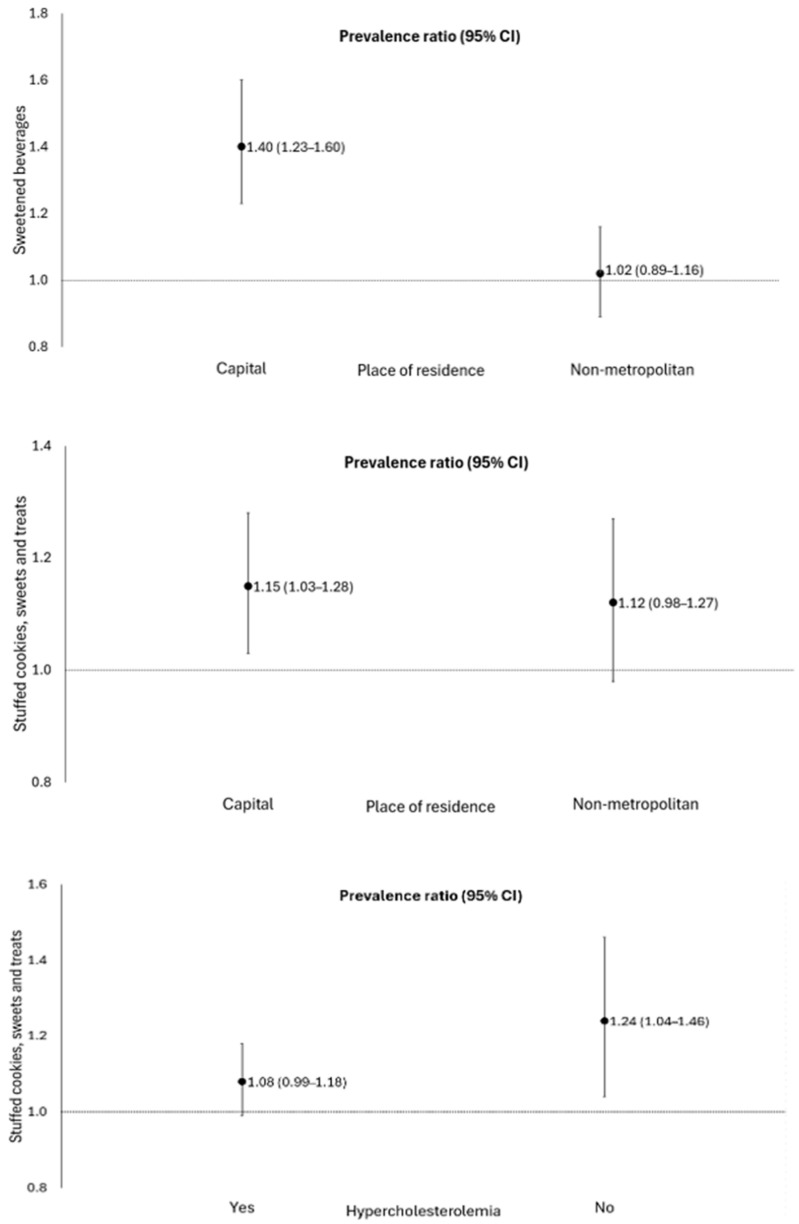
Association of the consumption of sweetened beverages and sandwich cookies, sweets, and treats with the occurrence of sequelae of COVID-19, stratified by place of residence and the presence of hypercholesterolemia, in elderly people. Roraima, Brazil, 2020. Poisson regression with robust variance, *p* < 0.05. Adjustment for sex, age, place of residence, education level, skin color/race, hospitalization, smoking, hypertension, diabetes *mellitus*, hypercholesterolemia, obesity, and chronic kidney disease.

**Table 1 idr-17-00025-t001:** Demographic characteristics according to the presence of sequelae of COVID-19 (PASC) in elderly people. Roraima, Brazil, 2020.

Variables		Post-Acute Sequelae of COVID-19		*p*-Value
	Total (%)	Yes (%)	No (%)	
**Sex**				
Male	45.0	58.7	41.3	<0.04
Female	55.0	64.2	35.8	
**Age range (years)** ^1^				
60–69	53.9	53.4	46.6	<0.001
70–79	31.5	69.7	30.3	
80 or more	14.7	75.3	24.7	
**Skin color/race**				
Yellow (Asian)	4.6	85.2	14.8	<0.001
White	22.8	58.6	41.4	
Brown	45.2	57.3	42.7	
Black	16.8	62.6	37.4	
Indigenous	10.6	75.7	24.3	
**Education level** ^1^				
No study	47.2	67.0	33.0	<0.001
Up to 8 years	29.9	60.0	40.0	
More than 8 years	22.9	53.1	46.9	
**Place of residence**				
Non-metropolitan	35.9	73.1	26.9	<0.001
Capital	64.1	55.4	44.6	

Pearson’s Chi-square test. ^1^ Chi-square test for linear trend.

**Table 2 idr-17-00025-t002:** Association and interaction between the consumption of healthy food groups and the occurrence of COVID-19 sequelae in elderly people. Roraima, Brazil, 2020.

Foods	Crude Model	Adjusted Model	Interactions	
	PR (95%CI)	PR (95%CI)	Covariates	PR (95%CI)
Beans	1.02 (0.92–1.12)	0.98 (0.90–1.06)	-	-

Fruits	0.81 (0.75–0.88)	**0.92 (0.85–0.99)**	Sex	0.94 (0.81–1.09)
			Age	**1.01 (1.002–1.02)**
			Place of residence (capital and non-metropolitan)	0.87 (0.76–1.01)
			Education level	1.11 (0.93–1.32)
			Skin color/race	0.95 (0.78–1.16)
			Hospitalization (yes/no)	**1.16 (1.02–1.32)**
			Hypertension	0.96 (0.70–1.32)
			Diabetes *mellitus*	1.19 (0.98–1.44)
			Hypercholesterolemia	1.05 (0.87–1.26)
			Obesity	1.05 (0.88–1.28)
			Chronic kidney disease (yes/no)	**1.17 (1.01–1.37)**
			Smoking	0.97 (0.82–1.16)

Vegetables	0.89 (0.81–0.97)	0.99 (0.92–1.07)	-	-

PR prevalence ratio; 95% CI, 95% confidence interval. Poisson regression adjusted for age, sex, education level, skin color, location, hospitalization, hypertension, diabetes mellitus, hypercholesterolemia, obesity, chronic kidney disease, and smoking. Bold values indicate statistical significance (*p* < 0.05).

**Table 3 idr-17-00025-t003:** Association and interaction between the consumption of markers of unhealthy food groups with the occurrence of COVID-19 sequelae in elderly people. Roraima, Brazil, 2020.

Foods	Raw Model	Adjusted Model	Interactions	
	PR (95%CI)	PR (95%CI)		PR (95%CI)
Burger and sausages	1.34 (1.24–1.46)	1.05 (0.97–1.13)	-	-

Sugar-sweetened beverages	1.56 (1.42–1.71)	**1.23 (1.12–1.35)**	Sex	0.92 (0.77–1.10)
			Age	0.99 (0.98–1.00)
			Place of residence (capital and non-metropolitan)	**1.53 (1.29–1.82)**
			Education level	1.06 (0.79–1.23)
			Skin color/race	0.86 (0.69–1.07)
			Hospitalization	0.99 (0.82–1.21)
			Hypertension	0.92 (0.68–1.24)
			Diabetes *mellitus*	0.88 (0.72–1.07)
			Hypercholesterolemia	0.85 (0.70–1.04)
			Obesity	0.88 (0.71–1.09)
			Chronic kidney disease	0.97 (0.79–1.21)
			Smoking	0.90 (0.69–1.17)

Instant noodles, packaged snacks or savory crackers	1.36 (1.26–1.48)	1.04 (0.96–1.12)	-	-

Sandwich cookies, sweets and treats	1.42 (1.30–1.55)	**1.12 (1.03–1.22)**	Sex	1.07 (0.91–1.25)
			Age	0.99 (0.99–1.00)
			Place of residence (capital and non-metropolitan)	**1.17 (1.003–1.38)**
			Education level	0.99 (0.82–1.18)
			Skin color/race	1.08 (0.89–1.31)
			Hospitalization	0.98 (0.83–1.14)
			Hypertension	0.98 (0.73–1.32)
			Diabetes *mellitus*	0.99 (0.82–1.21)
			Hypercholesterolemia (yes/no)	**0.82 (0.68–0.99)**
			Obesity	1.10 (0.88–1.37)
			Chronic kidney disease	0.85 (0.71–1.01)
			Smoking	0.92 (0.75–1.11)

PR prevalence ratio; 95% CI, 95% confidence interval. Poisson regression model adjusted for age, sex, Place of residence, education level, skin color/race, hospitalization, hypertension, diabetes *mellitus*, hypercholesterolemia, obesity, chronic kidney disease, and smoking. Bold values indicate statistical significance (*p* < 0.05).

## Data Availability

All data used and/or analyzed during the current study are available from the corresponding authors upon reasonable request.

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
