# Peer review of "Unhealthy Food Consumption Is Associated with Post-Acute Sequelae of COVID-19 in Brazilian Elderly People"

_2036-7449, 2025, doi:10.3390/idr17020025_

Round 1
Reviewer 1 Report
Comments and Suggestions for Authors
- Line 88-90: the sentence “Individuals with….pandemic” sounds an incomplete sentence. Would you consider to add text e.g. “…were included”
- Line 95: Punctuation is missing after … Health ”. Given…
- Line 107: is this paragraph referring to the pilot or the initial data collection (i.e. by DVE)? pls clarify
- Line 129: could you indicate which assessment method is used in the context of the “food consumption marker form”. Is it a modified 24hour recall or a food frequency quest, or another approach?
- Lines 133-138: pls shortly explain the reasonable for defining the particular food groups to estimating food consumption.
- Line 167: what do you mean with the term “interior” in this case, “indoors”? Or compared to “capital” in table1 (i.e. a rural area), you mean people leaving in urban areas?
- Line 282: since the food additives are officially approved ones, could their suspected trigger on inflammation cause generally a negative impact on health, and is that raising concerns?
- Line 292: is there a low food supply in these neighborhoods or are the elderlies buying food they can afford i.e. of lower quality?
- Line 299: “…and in Italy” or “…in an Italian study”

some minor corrections needed
Author Response
Response Letter to the Reviewer
Dear Editor,
We would like to thank you for the opportunity to revise and improve our manuscript entitled "Unhealthy food consumption is associated with post-acute sequelae of COVID-19 in Brazilian elderly people" (ID: idr-3444990). We also appreciate the reviewers for their constructive comments and suggestions, which significantly contributed to the improvement of our work.
Below, we respond in detail to each of the reviewers' comments:
Reviewer 1
Comment 1: Line 88-90: the sentence “Individuals with….pandemic” sounds an incomplete sentence. Would you consider to add text e.g. “…were included”
Response: We thank the reviewer for the suggestion. Following the recommendation, we have incorporated the phrase "were included" into the text to enhance clarity and precision in the wording.
Comment 2: Line 95: Punctuation is missing after … Health ”. Given…
Response: We thank the reviewer for the observation. We have followed the request and incorporated the punctuation after the word "health."
Comment 3: Line 107: is this paragraph referring to the pilot or the initial data collection (i.e. by DVE)? pls clarify
Response: We thank the reviewer for the question. We made the necessary adjustments to the text: With the initial data collection carried out by the DVE.
Comment 4: Line 129: could you indicate which assessment method is used in the context of the “food consumption marker form”. Is it a modified 24hour recall or a food frequency quest, or another approach?
Response: We thank the reviewer for the suggestion. We made the necessary adjustments to the text: Based on a quick and simplified screening questionnaire to assess the population's diet in all basic health units in the country.
Comment 5: Lines 133-138: pls shortly explain the reasonable for defining the particular food groups to estimating food consumption.
Response: We thank the reviewer for the suggestion. We have incorporated the following justification for the choice of food groups: These groups were selected for their ability to reflect diet quality and health impact, based on nutritional guidelines and scientific evidence
Comment 6: Line 167: what do you mean with the term “interior” in this case, “indoors”? Or compared to “capital” in table1 (i.e. a rural area), you mean people leaving in urban areas?
Response: We thank the reviewer for the question. In this case, the term 'interior' refers to non-metropolitan areas, meaning regions outside the capital, which may include both rural and urban areas that are not part of the capital city. Therefore, the text has been corrected to "non-metropolitan."
Comment 7: Line 282: since the food additives are officially approved ones, could their suspected trigger on inflammation cause generally a negative impact on health, and is that raising concerns?
Response: We thank the reviewer for the question. Although these additives are approved for use in the food industry, evidence suggests that prolonged exposure may contribute to low-grade systemic inflammation, raising concerns about their long-term health effects, especially in individuals with pre-existing conditions.
Comment 8: Line 292: is there a low food supply in these neighborhoods or are the elderlies buying food they can afford i.e. of lower quality?
Response: We thank the reviewer for the question. The lower income and education levels of older adults may worsen this situation because living in neighborhoods with a low supply of healthy foods often forces them to purchase more affordable but less healthy foods. The combination of limited availability and financial constraints leads to greater consumption of unhealthy foods
Comment 9: Line 299: “…and in Italy” or “…in an Italian study”
Response: We thank the reviewer for the observation. We have made the necessary corrections to the text: “and Italy”
In addition to the proposed revisions, we have refined the English to enhance the clarity and comprehensibility of the text.

Reviewer 2 Report
Comments and Suggestions for Authors
It was a very interesting study. The sample size is large. But it is one time study. The food consumption assessment was done by simply probing the question of ''did you eat this yesterday?" By doing so, it is difficult to understand how much and how often the patient consumed that specific foods. Why did you prefer to use poison regression for your analysis?
Comments on the Quality of English LanguageThe language needs improvement. I observed many errors which require editorial corrections.
Author Response
Response Letter to the Reviewer
Dear Editor,
We would like to thank you for the opportunity to revise and improve our manuscript entitled "Unhealthy food consumption is associated with post-acute sequelae of COVID-19 in Brazilian elderly people" (ID: idr-3444990). We also appreciate the reviewers for their constructive comments and suggestions, which significantly contributed to the improvement of our work.
Below, we respond in detail to each of the reviewers' comments:
Reviewer 2
We thank the reviewer for the comment. Additional information about the instrument used has been added to provide clearer context regarding the methodology employed. Regarding the use of the statistical test, poisson regression with robust variance may be preferable to logistic regression, since its coefficient estimates are interpreted as prevalence ratios (for cross-sectional studies), which may be more intuitive than the odds ratios estimated in binary logistic regression. Furthermore, the use of odds ratios may overestimate associations in cross-sectional studies, especially when the outcome assessed has a high prevalence (>10%), as investigated in this study 61,7%.
In addition to the proposed revisions, we have refined the English to enhance the clarity and comprehensibility of the text.
References:
Barros, A. J., & Hirakata, V. N. (2003). Alternatives for logistic regression in cross-sectional studies: an empirical comparison of models that directly estimate the prevalence ratio. BMC medical research methodology, 3, 21. https://doi.org/10.1186/1471-2288-3-21
Chen, W., Qian, L., Shi, J., & Franklin, M. (2018). Comparing performance between log-binomial and robust Poisson regression models for estimating risk ratios under model misspecification. BMC medical research methodology, 18(1), 63. https://doi.org/10.1186/s12874-018-0519-5
Chen, W., Shi, J., Qian, L., & Azen, S. P. (2014). Comparison of robustness to outliers between robust poisson models and log-binomial models when estimating relative risks for common binary outcomes: a simulation study. BMC medical research methodology, 14, 82. https://doi.org/10.1186/1471-2288-14-82
Talbot, D., Mésidor, M., Chiu, Y., Simard, M., & Sirois, C. (2023). An Alternative Perspective on the Robust Poisson Method for Estimating Risk or Prevalence Ratios. Epidemiology (Cambridge, Mass.), 34(1), 1–7. https://doi.org/10.1097/EDE.0000000000001544
